# Learnings from Scaling Visual Tokenizers for Reconstruction and Generation

**Philippe Hansen-Estruch** [1,2]  **David Yan** [2]  **Ching-Yao Chuang** [2]  **Orr Zohar** [2,3]  **Jialiang Wang** [2]  **Tingbo Hou** [2]
**Tao Xu** [2]  **Sriram Vishwanath** [1]  **Peter Vajda** [2]  **Xinlei Chen** [4]

## Abstract

Visual tokenization through auto-encoding enhances state-of-the-art image and video generative models by compressing pixels into a latent space. However, questions persist regarding how the design of the auto-encoder affects both reconstruction and downstream generative performance. This paper investigates the impact of scaling auto-encoders for reconstruction and generation by substituting the convolutional backbone with an enhanced Vision Transformer for Tokenization (ViTok). This paper's results show that scaling the auto-encoder bottleneck correlates with improved reconstruction, though its relationship with generative performance is more complex. In contrast, scaling the encoder does not lead to gains, while scaling the decoder enhances reconstruction with minimal effect on generation. These findings indicate that scaling the existing auto-encoder paradigm does not significantly improve generative performance. When paired with Diffusion Transformers, ViTok achieves competitive image reconstruction & generation performance on 256p and 512p ImageNet-1K. For videos, ViTok achieves state-of-the-art in both reconstruction & generation performance on 128p UCF-101.

## 1. Introduction

Modern methods for high-fidelity image and video generation (Brooks et al., 2024; Polyak et al., 2024; Genmo, 2024; Esser et al., 2024) rely on two components: a visual tokenizer that encodes pixels into a lower-dimensional latent space and subsequently decodes, and a generator that models this latent representation. While many studies have improved generators through scaling of Transformer-based architectures (Vaswani et al., 2017; Dosovitskiy et al., 2021),

the tokenizers themselves, predominantly based on convolutional neural networks (LeCun et al., 1998) (CNN), have rarely been the main focus of scaling efforts.

In this paper, we explore whether visual tokenizers require the same scaling efforts as generators. To do this, we first examine two potential bottlenecks: model architecture and data. We begin by replacing convolutional backbones with a Transformer-based auto-encoder (Vaswani et al., 2017), specifically adopting the Vision Transformer (ViT) (Dosovitskiy et al., 2021) architecture enhanced with Llama (Touvron et al., 2023), which has demonstrated effectiveness at scale (Gu & Dao, 2023; Sun et al., 2024). Our resulting auto-encoder design, which we refer to as *Vision Transformer Tokenizer* or *ViTok*, performs well with the generative pipeline in Diffusion Transformers (DiT) (Peebles & Xie, 2023). Second, we train our models on large-scale image datasets that significantly exceed ImageNet-1K (Deng et al., 2009) and extend our approach to videos, ensuring that our tokenizer scaling is not constrained by data limitations. Within this framework, we investigate three aspects:

- **Scaling the auto-encoding bottleneck.** Bottleneck size correlates with reconstruction metrics. However, when the bottleneck becomes large, the generative performance decreases due to increased channel sizes.

- **Scaling the encoder.** Scaling the encoder does not improve the generation outcomes and can even be detrimental. In particular, more complex latents can be more difficult to decode and model, reducing overall performance.

- **Scaling the decoder.** Scaling the decoder improves reconstruction quality, but its influence on downstream generative tasks remains mixed. We hypothesize that the decoder acts in part as a generator, filling in local textures based on limited information.

Collectively, these results indicate that scaling the auto-encoder tokenizer alone is not an effective strategy to enhance generative metrics within the current auto-encoding paradigm (Esser et al., 2021). We also observe that similar bottleneck trends apply to video tokenizers. However, ViTok leverages the inherent redundancy in video data more

---

[1]UT Austin [2]GenAI, Meta [3]Stanford University [4]Fundamental AI Research, Meta. Correspondence to: Philippe Hansen-Estruch <philippehansen@utexas.edu>.

*Proceedings of the 42nd International Conference on Machine Learning*, Vancouver, Canada. PMLR 267, 2025. Copyright 2025 by the author(s).

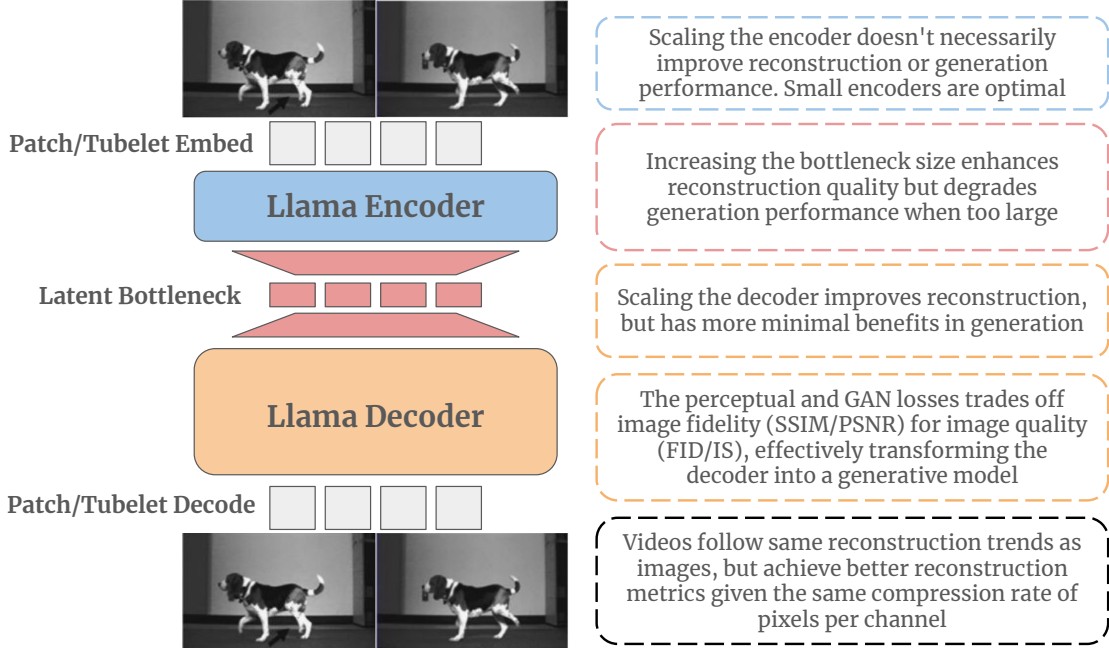

*Figure 1.* **Our learnings from scaling ViTok.** We present ViTok, an asymmetric auto-encoder combining Vision Transformers (ViTs) and an enhanced Llama architecture for reconstruction and generation. Visual inputs are embedded as patches/tubelets, processed by a compact Llama Encoder, and bottlenecked into latent codes. A larger Llama Decoder reconstructs the input. We detail findings of scaling the encoder, adjusting the bottleneck size, and expanding the decoder on the right.

effectively, achieving superior reconstruction metrics than for images at a fixed compression rate of pixels per channel. We summarize our findings and our method in Figure 1.

We compare our best-performing tokenizers against prior state-of-the-art methods. ViTok achieves image reconstruction and generation performance at 256p and 512p resolutions that matches or surpasses current state-of-the-art tokenizers on the ImageNet-1K (Deng et al., 2009) and COCO (Lin et al., 2014) datasets, all while utilizing 2–5× fewer FLOPs. In video applications, ViTok surpasses current state-of-the-art methods, achieving state-of-the-art results in 16-frame 128p video reconstruction and class-conditional video generation on the UCF-101 (Soomro, 2012) dataset.

## 2. Background

We review background on continuous visual tokenizers and then describe ViTok to enable our exploration.

### 2.1. Continuous Visual Tokenization

The Variational Auto-Encoder (VAE) (Kingma & Welling, 2013) is a framework that takes a visual input $X \in \mathbb{R}^{T \times H \times W \times 3}$ (where $T = 1$ for images and $T > 1$ for videos) is processed by an encoder $f_\theta$, parameterized by $\theta$. This encoder performs a spatial-temporal downsampling by a factor of $q \times p \times p$, producing a latent code. The

encoder outputs parameters for a multivariate Gaussian distribution—mean $z_m$ and variance $z_v$ with $c$ channel size.:

$$z \sim \mathcal{N}(z_m, z_v) = Z = f_\theta(X) \in \mathbb{R}^{\frac{T}{q} \times \frac{H}{p} \times \frac{W}{p} \times c}$$

The sampled latent vector $z$ is then fed into a decoder $g_\psi$, with parameters $\psi$, which reconstructs the input image $\hat{X} = g_\psi(z)$. The primary objective of the auto-encoder is to minimize the mean squared error between the reconstructed and original images, $\mathcal{L}_{\text{REC}}(\hat{X}, X)$. To regularize the latent distribution to a unit Gaussian prior which is necessary to recover the variational lower bound, a KL divergence regularization term is added, which we refer to as $\mathcal{L}_{\text{KL}}$. Recent advancements in VAEs used for downstream generation tasks (Esser et al., 2021; Rombach et al., 2022) incorporate additional objectives to improve the visual fidelity of the reconstructions. These include a perceptual loss based on VGG features (Johnson et al., 2016) $\mathcal{L}_{\text{LPIPS}}$ and an adversarial GAN objective, $\mathcal{L}_{\text{GAN}}$ (Goodfellow et al., 2014). The comprehensive loss function for the auto-encoder, $\mathcal{L}_{\text{AE}}(\hat{X}, X, Z)$, is formulated as:

$$\mathcal{L}_{\text{AE}}(\hat{X}, X, Z) = \mathcal{L}_{\text{REC}}(\hat{X}, X) + \beta \mathcal{L}_{\text{KL}}(Z) \\ + \eta \mathcal{L}_{\text{LPIPS}}(\hat{X}, X) + \lambda \mathcal{L}_{\text{GAN}}(\hat{X}, X) \quad (1)$$

where $\beta$, $\eta$, and $\lambda$ are parameters that control the trade-off between visual fidelity and quality (Section 3.4).

Table 1. **Model Sizes and FLOPs for ViTok.** We describe ViTok variants by specifying the encoder and decoder sizes separately. For example, ViTok S-B/4x16 refers to a model with an encoder Small (S) and a decoder Base (B), using tubelet size $q = 4$ and $p = 16$.

| Model | Hidden Dimension | Blocks | Heads | Parameters (M) | GFLOPs |
|---|---|---|---|---|---|
| Small (S) | 768 | 6 | 12 | 43.3 | 11.6 |
| Base (B) | 768 | 12 | 12 | 85.8 | 23.1 |
| Large (L) | 1152 | 24 | 16 | 383.7 | 101.8 |

## 2.2. Scalable Auto-Encoding Framework

We develop a visual tokenizer based on a Vision Transformer (ViT) architecture, replacing CNNs for better scalability. Building on ViViT (Arnab et al., 2021), our framework handles both images and videos. A 3D convolution with kernel and stride size $q \times p \times p$ tokenizes the input $X$ into $X_{\text{embed}} \in \mathbb{R}^{B \times L \times C_f}$, where $L = \frac{T}{q} \times \frac{H}{p} \times \frac{W}{p}$. A ViT encoder processes $X_{\text{embed}}$, and a linear projection produces a compact representation $Z = f_\theta(X_{\text{embed}}) \in \mathbb{R}^{B \times L \times 2c}$. Following the VAE formulation, we recover $z \in \mathbb{R}^{B \times L \times c}$ and define the latent space dimensionality as

$$E = L \times c, \tag{2}$$

which controls the compression ratio. The decoder upsamples $z$ from $c$ to $C_g$ channels via a linear projection, processes it with a ViT decoder, and uses a 3D transposed convolution to recover $\hat{X}$. This defines our *Vision Transformer Tokenizer* (ViTok). Figure 1 illustrates the process, and Table 1 provides ViTok size details for reference.

## 2.3. Experiment Setup and Training

To address instability in VAE frameworks, we use a two-stage training approach. Stage 1 trains with MSE, LPIPS, and KL losses ($\beta = 1 \times 10^{-3}$, $\eta = 1.0$, $\lambda = 0$) for stable auto-encoding. Stage 2 incorporates the GAN, freezes the encoder, and fine-tunes the decoder with $\lambda = 1.0$.

**Architecture, datasets, and training details.** We use a Vision Transformer architecture (ViT) for the encoder and decoder, incorporating SwiGLU and 3D Axial RoPE for spatiotemporal modeling. We train on large-scale datasets: Shutterstock (450M images) and ImageNet-1K for images, and Shutterstock videos (30M videos) for video. Evaluation is performed on ImageNet-1K, COCO-2017, UCF-101, and Kinetics-700. Stage 1 runs for 100k steps with batch sizes of 1024 (images) and 256 (videos). Stage 2 fine-tunes for another 100k steps. We use AdamW ($\beta_1 = 0.9$, $\beta_2 = 0.95$), a peak learning rate of $\frac{1 \times 10^{-4}}{256}$, weight decay of $1 \times 10^{-4}$, and a cosine decay schedule. For Stage 2, we use Style-GAN (Karras et al., 2019) discriminator with a learning rate of $2 \times 10^{-5}$ and a 25k-step warmup. Training uses bfloat16 autocasting, with EMA (0.9999) introduced in Stage 2.

**Reconstruction evaluation metrics.** To gauge reconstruction quality, we use the Fréchet Inception Distance (FID) (Heusel et al., 2017), Inception Score (IS) (Salimans et al., 2016), Structural Similarity Index Measure (SSIM) (Wang et al., 2004), and Peak Signal-to-Noise Ratio (PSNR). For video, we report rFID (frame-wise FID) and Fréchet Video Distance (rFVD) (Unterthiner et al., 2019).

**Generation experiments and metrics.** To assess our tokenizers in a large-scale generative setting, we train a class-conditional DiT-L (Peebles & Xie, 2023) with 400M parameters for 500,000 steps and a batch size of 256, applying classifier-free guidance (CFG) (Ho & Salimans, 2022) on a DDIM sampler (Song et al., 2020) over 250 steps and a CFG scale of 1.5. We apply the same Llama upgrades to our DiT as for our tokenizers. We measure generation quality using gFID and gIS (gInception Score) calculated over 50,000 samples. Since ViTok can directly output continuous tokens, we can feed the noised latents $z + \epsilon$ directly into DiT without patchifying and predict the noise.

## 3. Bottlenecks, Scaling, and Trade-offs in Visual Tokenization

In Section 2, we introduced ViTok and its training process. Here, we explore the impact of scaling three key factors—bottleneck size, encoder size, and decoder size—on reconstruction and generation performance.

### 3.1. $E$ as the Main Bottleneck in Image Reconstruction

In prior discrete cases performance depends on the tokens ($L$) and the codebook size (Oord et al., 2017; Mentzer et al., 2023). For ViTok, the analogous factor is $E$ (Equation 2), which proves to be the critical determinant of reconstruction performance. As described in Equation 2, $E$ represents the total number of floating points (also dependent on precision) and relates to the compression ratio, defined as $\frac{T \times H \times W \times 3}{E}$. We train ViTok S-B on stage 1 (Section 2.3) on 256p image reconstruction, sweeping patch sizes $p = \{32, 16, 8\}$ and channel widths $c = \{4, 8, 16, 32, 64\}$, yielding $E$ values from $2^8$ to $2^{16}$. Patch size affects $L = \frac{H \times W}{p^2}$ and therefore FLOPs due to the quadratic nature of attention, while $c$ controls the bottleneck between encoder and decoder. The results are summarized in Figure 3, with more details of the experiment provided in Appendix A.

Figure 3 shows a strong correlation between $E$ and recon-

| Ground Truth | Reconstruction with $E$ Floating Points | | | | |
| --- | --- | --- | --- | --- | --- |
| | 16384 | 8192 | 4096 | 2048 | 1024 |

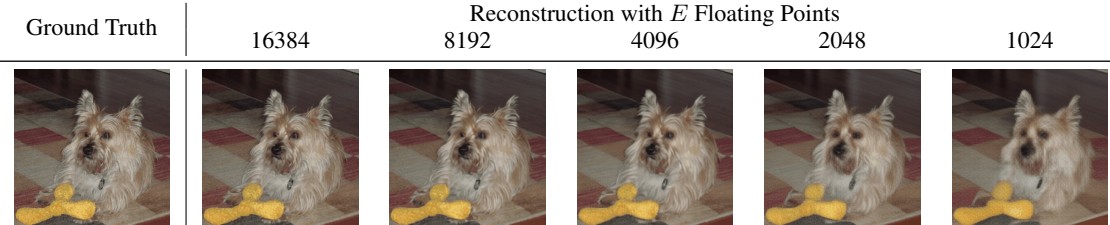

*Figure 2.* **256p image reconstruction visualization over floating points $E$.** Example reconstruction for varying $E$ values on ViTok S-B/16, achieved by adjusting the channel size $c = 64, 32, 16, 8, 4$ for each image across the row. As $E$ decreases, high-frequency details diminish, with small colors and fine details gradually lost. When $E < 4096$, textures merge, and significant detail loss becomes apparent.

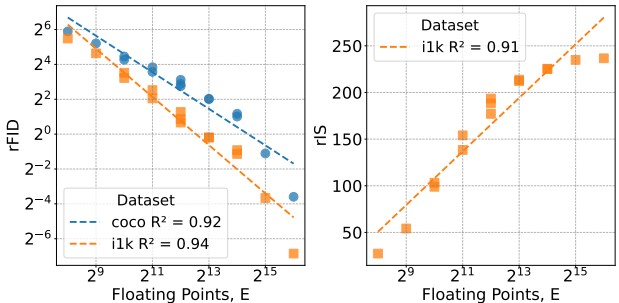

*Figure 3.* **256p image reconstruction sweep over floating points $E$.** We evaluate ViTok S-B using combinations of patch sizes $p \in 8, 16, 32$ and channel widths $c \in 4, 8, 16, 32, 64$ to investigate how $E = \frac{256^2}{p^2} \cdot c$ influences FID and IS in reconstruction tasks. This indicates that $E$ is the primary bottleneck for reconstruction, irrespective of the code shape or FLOPs expended.

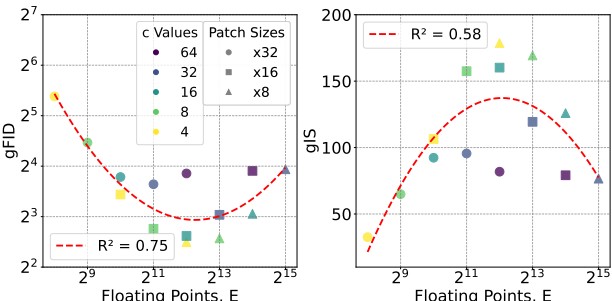

*Figure 4.* **256p image generation over $E$.** We evaluate each tokenizer from Figure 3 on DiT following Section 2.3. Our results show no strong linear correlation between $\log(E)$ and generation performance. Instead, a second-order trend reveals an optimal $E$ for each patch size $p$. This highlights the necessity of optimizing both $E$ and $c$ to balance reconstruction with generation.

struction metrics (rFID/rIS), indicating that $E$ is a key predictor of reconstruction quality. Performance trends are consistent across datasets, with minor rFID variations due to validation set sizes (50k for ImageNet-1K vs 5k for COCO). For a fixed $E$, varying patch sizes ($c = \frac{E \times p^2}{H \times W}$) yields a similar performance, suggesting that FLOPs do not improve results for a given $E$. This establishes $E$ as the primary bottleneck. Figure 2 visualizes reconstructions for different $E$ values. As $E$ decreases, high-frequency details are lost; for $E < 4096$, textures degrade significantly.

> **Finding 1:** Regardless of code shape or flops expended in auto-encoding, the total number of floating points in the latent code ($E$) is the most predictive bottleneck for visual reconstruction.

### 3.2. The Impact of $E$ in Image Generation

In this section, we explore how $E$ impacts generative performance using the tokenizers from Figure 3 and the training protocol in Section 2.3. Results are shown in Figure 4.

Unlike reconstruction, the generative metrics ($\log(\text{gFID})$ and gIS) do not show a linear correlation with $\log(E)$. Instead, each patch size has an optimal $E$: $p = 16, c = 16$,

$E = 4096$; $p = 8, c = 4, E = 4096$; and $p = 32, c = 32$, $E = 2048$. In particular, gFID and gIS remain poor for $c > 32$, indicating that excessive $c$ harms performance.

Low $E$ bottlenecks the generative model due to poor reconstruction, while high $E$ (driven by large $c$) complicates convergence and degrades gFID and gIS. This aligns with recent findings on the trade-off between rFID and gFID in latent diffusion models (Yao & Wang, 2025). The balance of $E$ and $c$ is crucial: they must be low enough to aid generation, but high enough to ensure quality reconstructions. Generation visualizations are provided in the Appendix B.

> **Finding 2:** Scaling $E$ does not consistently improve generative performance. Instead, optimal results are achieved by tuning both $E$ and $c$. A low $E$ limits the quality of reconstruction, while a high $E$ and channel size $c$ hinder the generation performance.

### 3.3. Scaling Trends in Auto-Encoding

We explore how scaling impacts auto-encoding in reconstruction and generation tasks using ViTok, fixing parameters to $p = 16, c = 16$. We sweep sizes S-S, B-S, S-B, B-B,

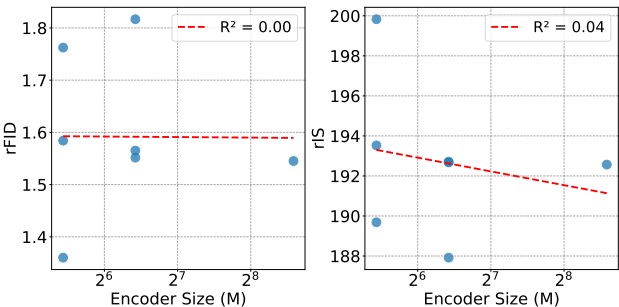

Figure 5. **Encoder scaling on 256p image reconstruction.** We evaluate reconstruction metrics of ViTok over model sizes S-S, B-S, S-B, B-B, B-L, L-L with fixed $p = 16, c = 16$. There is no correlation between encoder size and reconstruction performance, indicating that scaling the encoder is unhelpful.

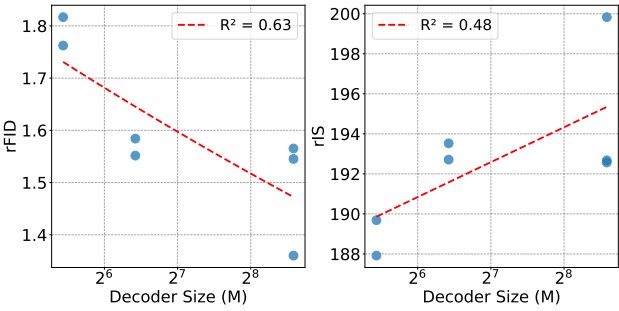

Figure 6. **Decoder scaling on 256p image reconstruction.** Using the results from Figure 5, we plot various decoder sizes (S, B, L) over reconstruction performance. There is a strong correlation between decoder size and reconstruction performance, which indicates scaling the decoder improves reconstruction. Although, increasing the decoder size from Base to Large does not provide the same boost of performance as doubling $E$ to 8192 from 4096.

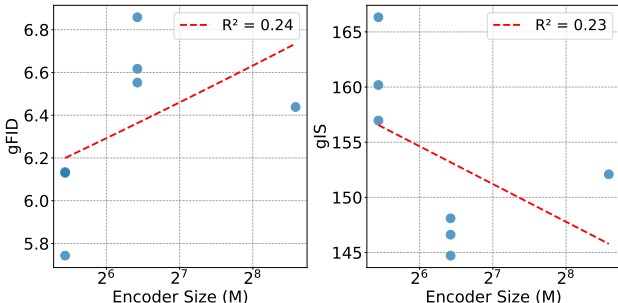

Figure 7. **Encoder scaling on 256p image generation.** We evaluate each tokenizer from Figure 5 on DiT following Section 2.3. We plot encoder size over generation metric results. There is a weak negative correlation between encoder size and final performance indicating that scaling the encoder is harmful for generation.

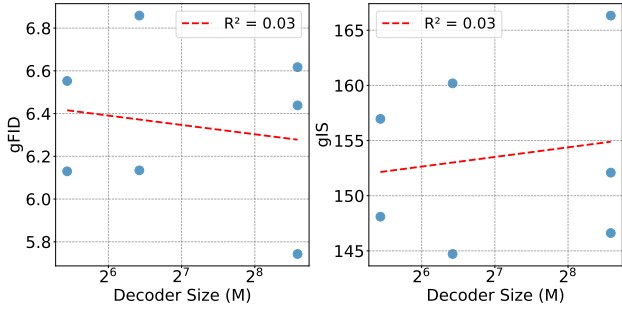

Figure 8. **Decoder scaling on 256p image generation.** Using the results from Figure 6, we plot various decoder sizes (S, B, L) over generation performance. We plot decoder size over generation metric results for CFG scales of 1.5. Unlike reconstruction, there is no correlation between decoder size and generation performance. This indicates that scaling the decoder has minimal benefits overall and most scaling efforts should focus on the generator.

S-L, B-L, L-L, defined in Table 1 following the training protocol in Section 2.3. Results are in Figures 5 and 6.

Figure 5 shows no correlation between encoder size and reconstruction performance. In contrast, Figure 6 shows a correlation between decoder size and reconstruction quality. However, $E$ remains the dominant factor: Going from Base to Large reduces rFID from 1.6 to 1.3 for $E = 4096$, while doubling $E$ to 8192 with a Base decoder drops rFID to 0.8. Scaling the decoder can be beneficial, but scaling the encoder provides no significant advantage.

Figures 7 and 8 examine the impact of scaling the encoder and decoder on generation performance. Figure 7 shows a slight negative correlation between encoder size and generation quality, suggesting that larger encoders either have no benefit or may harm performance. Similarly, Figure 8 indicates that scaling the decoder offers minimal gains for generation. Unlike reconstruction, scaling the encoder or decoder does not significantly improve generation quality

but raises training and inference costs.

> **Finding 3:** Scaling the encoder provides no reconstruction benefits and may harm generation, while scaling the decoder improves reconstruction but offers limited generation gains.

*With the findings so far, we believe simply scaling the current auto-encoding (Esser et al., 2021) based tokenizers does not automatically lead to improved downstream generation performance. Therefore, for generation tasks, it is more cost-effective to concentrate scaling efforts on the generator itself, rather than the visual tokenizer.*

### 3.4. A Trade-Off in Decoding

We compare the manner in which different losses balance SSIM/PSNR against FID. SSIM/PSNR measure visual fidelity or how much of the original information is preserved, while FID focus on visual quality and how closely outputs

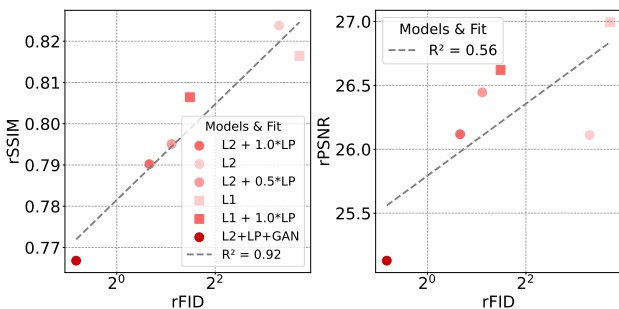

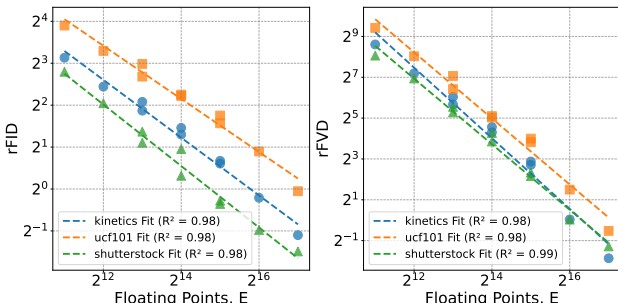

*Figure 9.* **Metric trade-offs in 256p image reconstruction.** We train ViTok S-B/16 varying the LPIPS (LP in figure) weight $\lambda \in \{0.0, 0.5, 1.0\}$ and using either L1 or L2 MSE reconstruction loss (Equation 1). Additionally, we finetune ViTok S-B/16 with GAN and include the result as L2+LP+GAN. The results indicate that enhancing rFID scores through increased perceptual and visual losses requires a trade-off with rSSIM/rPSNR, resulting in loss of information from the original image.

*Figure 10.* **256p video reconstruction results over $E$.** We train Vi-Tok S-B on $16 \times 256 \times 256$ videos at 8 fps, varying $p \in \{8, 16, 32\}$ and $q \in \{1, 2, 4, 8\}$ with $c = 16$. Reconstruction performance is evaluated using rFID and rFVD on the Kinetics-700, UCF101, and Shutterstock datasets. The results exhibit a similar trend to image reconstruction in Figure 3, demonstrating a strong correlation between $E$ and reconstruction performance. Expectantly, videos are more compressible than a direct scaling from images would suggest; instead of requiring $16 \times E$, achieving comparable rFID to 256p image reconstruction only necessitates $4$–$8 \times E$.

match the real dataset. This comparison shows how different losses choices can change the role of the decoder from strictly reconstructing to more actively generating content.

We conducted these experiments on ViTok by fixing $p = 16$, $c = 16$, and $E = 4096$. We then trained with stage 1 and varied the LPIPS loss weight $\lambda \in \{0.0, 0.5, 1.0\}$ combined with the choice of L1 or L2 reconstruction loss (Equation 1). We also include our Stage 2 results following Section 2.3 to see the final effect of the generative adversarial objective.

Figure 9 shows a clear trade-off among these losses. Without perceptual loss, we get worse rFID scores but better rSSIM/rPSNR, indicating that a strict MSE-based approach preserves the most original information. Increasing $\lambda$ gradually lowers SSIM/PSNR while improving FID/IS. Finally, fine-tuning the decoder with a GAN pushes these generative metrics further, achieving an rFID of 0.50 at the cost of a lower SSIM/PSNR. These results demonstrate that at a fixed $E$, aiming for higher visual quality requires sacrificing some traditional compression fidelity.

> **Finding 4:** There is a trade-off between rFID (visual quality) and rSSIM/rPSNR (visual fidelity), influenced by the weights of Equation 1. Consequently, the decoder can be trained as a generation model, which extends the main generator.

### 3.5. Video Results

We extend ViTok to video tasks to study the impact of $E$ on video reconstruction and explore redundancy in video data. Using 16-frame videos at 8 fps and 256p resolution, we maintain consistency with image experiments. Furthermore, tokenizing videos can result in long sequences; for example, a tubelet size of 4×8 (temporal stride $q = 4$, spatial stride

$p = 8$) for a $16 \times 256 \times 256$ video yields 4096 tokens. Based on Section 3.3, we use a small ViTok S-B variant to reduce computational costs as scaling the encoder does not aid performance. We sweep patch sizes $p \in \{8, 16, 32\}$ and temporal strides $q \in \{1, 2, 4, 8\}$. Figure 10 shows that $\log(E)$ correlates with reconstruction metrics rFVD/rFID.

Videos require $E \approx 16384$ to achieve an rFID of 2.0, whereas images need $E = 4096$. The smaller-than-expected impact of scaling image-derived $E$ across video frames highlights ViTok's ability to exploit video compressibility.

> **Finding 5:** Videos share the same reconstruction bottleneck characteristics as images with respect to $E$. However, auto-encoding exploits video compressibility, allowing $E$ to scale more efficiently relative to total pixels than in images.

## 4. Experimental Comparison

In this section, we compare our auto-encoders to prior work on image reconstruction at 256p and 512p, as well as video reconstruction with 16 frames at 128p. We utilize the S-B/16 and S-L/16 variants for images and the S-B/4x8 variant for videos. Training these tokenizers follows the Stage 1 and Stage 2 protocol outlined in Section 2.3.

### 4.1. Image Reconstruction and Generation

We evaluate our models on image reconstruction and class-conditional generation using ImageNet-1K and COCO-2017 at 256p and 512p resolutions. For reconstruction, we compare against state-of-the-art continuous tokenizers, including SD-VAE 2.x (Rombach et al., 2022), SDXL-

*Table 2.* **256p image reconstruction comparison.** We assess the reconstruction performance of ViTok on ImageNet-1K and COCO-2017 validation sets, benchmarking them against CNN-based tokenizers with an equivalent compression ratio (48 pixels per channel). Our ViTok S-B/16 tokenizer achieves state-of-the-art (SOTA) rFID scores on both ImageNet-1K and COCO datasets, outperforming other CNN-based continuous tokenizers while utilizing significantly fewer FLOPs. Furthermore, ViTok maintains competitive performance in SSIM and PSNR metrics compared to prior methods. When scaling decoder size to Large, ViTok improves all its reconstruction numbers.

| Name | Params (M) | GFLOPs | ImageNet | | | COCO | | |
|---|---|---|---|---|---|---|---|---|
| | | | rFID↓ | rPSNR↑ | rSSIM↑ | rFID↓ | rPSNR↑ | rSSIM↑ |
| SD-VAE | 59.3 | 162.2 | 0.78 | 25.08 | 0.705 | 4.63 | 24.82 | 0.720 |
| SDXL-VAE | - | - | 0.68 | 26.04 | **0.834** | 4.07 | 25.76 | **0.845** |
| OAI | - | - | 0.81 | 24.43 | 0.786 | 4.59 | 24.19 | 0.800 |
| Cosmos-CI | - | - | 2.02 | **31.74** | 0.700 | 5.6 | **31.74** | 0.703 |
| ViTok S-B/16 | 129.0 | 34.8 | 0.50 | 24.36 | 0.747 | 3.94 | 24.45 | 0.759 |
| ViTok S-L/16 | 426.8 | 113.4 | **0.46** | 24.74 | 0.758 | **3.87** | 24.82 | 0.771 |

*Table 3.* **512p image reconstruction comparison.** We assess the reconstruction performance of our top-performing tokenizers on ImageNet-1K and COCO-2017 validation sets, benchmarking them against a CNN-based tokenizer with an equivalent compression ratio (48 pixels per channel). Our ViTok S-B/16 tokenizer maintains state-of-the-art (SOTA) results across all metrics.

| Name | Params(M) | GFLOPs | ImageNet | | | COCO | | |
|---|---|---|---|---|---|---|---|---|
| | | | rFID↓ | rPSNR↑ | rSSIM↑ | rFID↓ | rPSNR↑ | rSSIM↑ |
| SD-VAE | 59.3 | 653.8 | 0.19 | - | - | - | - | - |
| ViTok S-B/16 | 129.0 | 160.8 | **0.18** | 26.72 | 0.803 | **2.00** | 26.14 | 0.790 |

VAE (Podell et al., 2023), Consistency Decoder (OpenAI, 2023), and COSMOS (NVIDIA, 2024). Discrete tokenizers are excluded due to incompatibility with direct comparisons.

As shown in Table 2, our ViTok S-B/16 achieves SOTA rFID scores on ImageNet-1K and COCO, with competitive rSSIM and rPSNR metrics. Scaling the decoder to size L further improves performance. Notably, ViTok reduces FLOPs compared to prior CNN-based methods. For 512p reconstruction (Table 3), ViTok achieves SOTA performance with significantly fewer FLOPs, outperforming existing methods in both metrics and computational efficiency.

For class-conditional image generation, we train a DiT-XL (675M parameters) for 4M steps paired with ViTok S-B/16, using 256 tokens for 256p and 1024 tokens for 512p generation. Results in Table 5 show ViTok performs competitively against SD-VAE. At 512p, ViTok matches other methods, demonstrating its effectiveness at higher resolutions. Generated images are visualized in Figures 16 and 17.

### 4.2. Video Reconstruction and Generation

For our video comparison, our reconstruction metrics are computed on the UCF-101 training set and compared against state-of-the-art methods including TATS (Ge et al., 2022), LARP (Wang et al., 2024a), and MAGViTv1/v2 (Yu et al., 2023b;a). The results are presented in Table 4. Our tokenizers demonstrate very competitive performance relative to prior work. Specifically, S-B/4x8 (1024 tokens) achieves

state-of-the-art (SOTA) rFVD results compared to other CNN-based continuous tokenizers with the same total compression ratio. Additionally, our approach significantly reduces FLOPs compared to Transformer-based prior method LARP, underscoring the efficiency and versatility of ViTok.

We further evaluate our models on class-conditional video generation using the UCF-101 dataset. We train a DiT-L model for 500K steps on the UCF-101 training set, computing gFID and gFVD metrics. The results are summarized in Table 6. ViTok achieves SOTA gFVD scores at 1024 tokens. Example video generations using our 1024-token configuration are illustrated in Figure 21.

## 5. Related Work

**Image tokenization.** High-resolution images have been compressed using deep auto-encoders (Hinton et al., 2012; Vincent et al., 2008), a process that involves encoding an image into a lower-dimensional latent representation, which is then decoded to reconstruct the original image. Variational auto-encoders (VAEs) (Kingma & Welling, 2013) extend this concept by incorporating a probabilistic meaning to the encoding. VQVAEs (Oord et al., 2017) introduce a vector quantization (VQ) step in the bottleneck of the auto-encoder, which discretizes the latent space. Further enhancing the visual fidelity of reconstructions, VQGAN (Esser et al., 2021) integrates adversarial training into the objective of VQVAE. Finally, FSQ (Mentzer et al., 2023) simplifies the training

*Table 4.* **128p Video Reconstruction.** We evaluate ViTok S-B/4x8 on video reconstruction for $16 \times 128 \times 128$ video on UCF-101 11k train set and compare to prior that utilizes similar compression ratios. ViTok S-B/4x8 achieves SOTA performance in rFVD, rFVD, rPSNR, and rSSIM metrics compared to prior work. ViTok also reduces the total FLOPs required from prior transformer based methods.

| Method | Params(M) | GFLOPs | Compression | # Tokens | rFID↓ | rFVD↓ | rPSNR↑ |
| --- | --- | --- | --- | --- | --- | --- | --- |
| TATS | 32 | Unk | 2048 | - | 162 | - | - |
| MAGViT | 158 | Unk | 1280 | - | 25 | 22.0 | 0.701 |
| MAGViTv2 | 158 | Unk | 1280 | - | 16 | - | - |
| LARP-L-Long | 174 | 505.3 | 1024 | - | 20 | - | - |
| ViTok S-B/4x8 | 129 | 160.8 | 1024 | 2.13 | 8 | 30.11 | 0.923 |

*Table 5.* **Class conditional image generation results.** We evaluate our tokenizers on class-conditional generation at resolutions of 256p and 512p on ImageNet-1K compared to prior methods. ViTok performance is competitive with prior continuous diffusion generation methods for both 256p and 512p generation.

| Method | 256p Generation | | 512p Generation | |
| --- | --- | --- | --- | --- |
| | gFID↓ | gIS ↑ | gFID↓ | gIS ↑ |
| LDM | 3.60 | 247.70 | - | - |
| DiT-XL/2 | 2.27 | 278.24 | 3.04 | 240.82 |
| VQGAN | 15.78 | - | - | - |
| TiTok-B | 2.48 | - | 2.49 | - |
| ViTok S-B/16 | 2.45 | 284.39 | 3.41 | 251.46 |

*Table 6.* **128p class conditional video generation.** We evaluate ViTok S-B 4x8 on class-conditional generation $16 \times 128 \times 128$ on the UCF-101 dataset compared to prior methods. ViTok S-B/4x8 achieves SOTA video generation performance when used with a comparable compression ratio with prior methods.

| Tokenizer | Params | gFID↓ | gFVD↓ |
| --- | --- | --- | --- |
| TATS | 321M | - | 332 |
| MAGViT | 675M | - | 76 |
| MAGViTv2 | 177M | - | 58 |
| W.A.L.T | 177M | - | 46 |
| LARP-L-Long | 177M | - | 57 |
| ViTok S-B/4x8 | 400M | 6.67 | 27 |

process of VQVAE by avoiding additional auxiliary losses.

While ConvNets have traditionally been the backbone for auto-encoders, recent explorations have incorporated Vision Transformers (Vaswani et al., 2017; Kolesnikov et al., 2020) (ViT) to auto-encoding. UViM (Kolesnikov et al., 2022) adopts an asymmetric Vision Transformer (ViT) encoder–decoder to unify a wide range of vision tasks, including panoptic segmentation and depth prediction. ViT–VQGAN (Yu et al., 2022) replaces the convolutional auto-encoder of VQGAN with ViT blocks, yielding better reconstruction quality and more favorable scaling. TiTok (Yu et al., 2024) introduces a compact 1-D ViT tokenizer that distills latents from a VQGAN, freezes the encoder, and

fine-tunes only the decoder for downstream generative tasks. Causally Regularized Tokenization (CRT) (Ramanujan et al., 2024) and ElasticTok (Yan et al., 2024) both develop causal 1-D tokenizers tailored for autoregressive image models. Finally, JetFormer (Tschannen et al., 2025) unifies tokenization and generation by training an end-to-end autoregressive model that directly produces high-resolution images, removing the need for separate stage-1 and stage-2 components.

**Video tokenization.** VideoGPT (Yan et al., 2021) proposes using 3D Convolutions with a VQVAE. TATS (Ge et al., 2022) utilizes replicate padding to reduce temporal corruption issues with variable-length videos. Phenaki (Villegas et al., 2022) utilizes the Video Vision Transformer (Arnab et al., 2021)(ViViT) architecture with a factorized attention using full spatial and casual temporal attention. MAGViTv1 (Yu et al., 2023a;b) utilizes a 3D convolution with VQGAN to learn a video tokenizer coupled with a masked generative portion. Finally, LARP (Wang et al., 2024a) is a concurrent work that tokenizes videos with ViT into discrete codes similar to TiTok's architecture (Yu et al., 2024), our work differs as we use continuous codes and don't concatenate latent tokens to the encoder.

**High resolution generation.** High resolution image generation has been done prior from sampling VAEs, GANs (Goodfellow et al., 2014), and Diffusion Models (Sohl-Dickstein et al., 2015; Song & Ermon, 2019; Song et al., 2020; Ho et al., 2020). While some work perform image synthesis in pixel space (Dhariwal & Nichol, 2021), many works have found it more computationally effective to perform generation in a latent space from an auto-encoder (Rombach et al., 2022). Typically, the U-Net architecture (Ronneberger et al., 2015) has been used for diffusion modeling, although recently transformers have been gaining favor in image generation. MaskGIT (Chang et al., 2022) combines masking tokens with a schedule to generate images and Diffusion Transformers (Peebles & Xie, 2023) proposes to replace the U-Net architecture with a ViT. With Some methods use auto-regressive modeling to generate images (Ramesh et al., 2021; Yu et al., 2023a;b; Li

et al., 2024). DALL-E (Ramesh et al., 2021) encodes images with a VQVAE and then uses next token prediction to generate the images. While most auto-regressive image generators rely on discrete image spaces, MAR (Li et al., 2024) proposed a synergized next token predictor that allows for visual modeling in continuous latent spaces.

## 6. Conclusion

In this paper, we explored scaling in auto-encoders, introducing ViTok, a ViT-style auto-encoder. We investigated scaling bottleneck sizes, encoder sizes, and decoder sizes, finding a strong correlation between the total number of floating points ($E$) and visual quality metrics. Our results show that scaling the auto-encoder alone does not significantly improve generative performance. From our sweep, we develop SOTA visual tokenizers. ViTok achieves competitive performance with state-of-the-art tokenizers, matching rFID and rFVD metrics while using fewer FLOPs.

## 7. Limitations

While ViTok demonstrates strong performance, our findings—particularly the negative results—are constrained by the experimental setup and assumptions. These limitations are shared to prevent redundant efforts and to provide valuable insights for future work. Despite these constraints, the lessons learned offer contributions to understanding the core challenges in image and video generative methods.

## Impact Statement

This paper presents work whose goal is to advance the field of generative modeling and visual compression. There are many potential societal consequences of our work, none which we feel must be specifically highlighted here as our benchmarks are purely academic in nature.

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

## A. Extra Experiments

### A.1. Detailed 256p Image Results

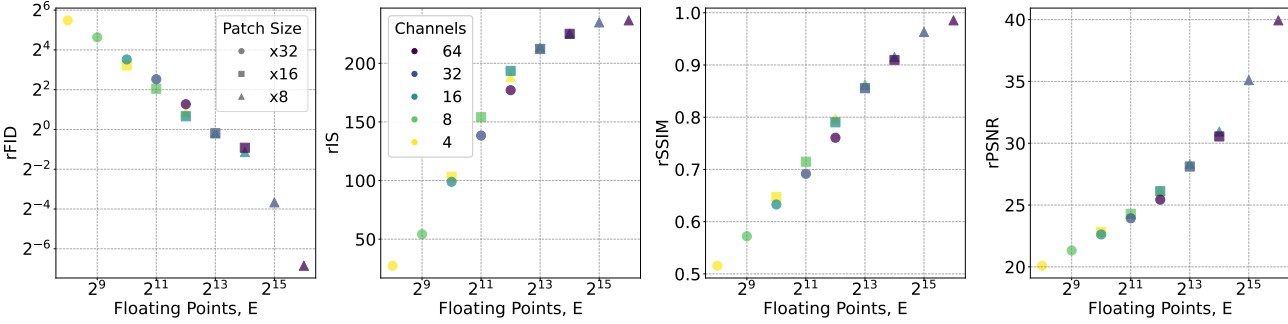

*Figure 11.* **256p Detailed Image Reconstruction Results with Fixed Architecture Size.** We provide more details for the sweep in Figure 3 on the just the ImageNet-1K validation set. For $1024 \leq E \leq 16384$, where intersections of $E$ exist across patch sizes, we see very little variation in performance for fixed $E$. This indicates that $E$ is the main bottleneck for visual auto-encoding and is not influence by increasing FLOPs.

We provide further detail of the ImageNet-1K validation reconstruction results from Figure 3 in Figure 11. Here we show different patch sizes and channels over $E$. This shows that regardless of patch size and FLOPs usage, $E$ is highly correlated with the reconstruction perforance

### A.2. $E$ trends on 512p reconstruction.

To examine how resolution size affects $E$, we scale up the resolution from 256p to 512p. We test ViTok S-B/16 over $p \in 8, 16, 32$. The results of the sweep are shown in Figure 12. The results follow a trend similar to that in Figure 3, with $E$ exhibiting consistent correlation relationships. While FID and IS are challenging to compare across resolutions[1], achieving comparable rSSIM and rPSNR performance at 512p requires $4 \times E$ from 256p. This suggests that maintaining performance across resolutions requires preserving the same compression ratio, $\frac{H \times W \times 3}{E}$.

---

[1]The InceptionV3 network used for FID and IS calculations resizes images to 299p before feature computation, leading to potential information loss during downsampling.

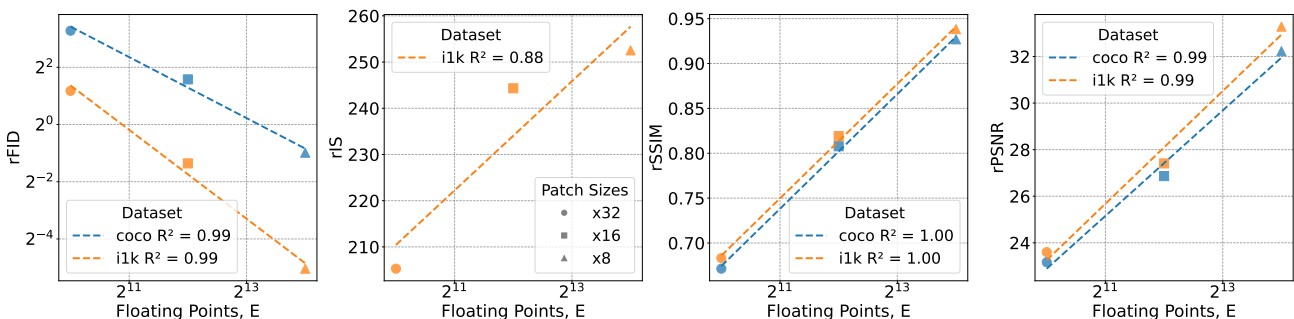

*Figure 12.* **512p Image reconstruction over** $E$. We evaluate ViTok S-B trained with stage 1 (Section 2.3) across all combinations of patch sizes $p \in 8, 16, 32$ and a fixed channel width $c = 16$, analyzing how the total floating-point operations, calculated as $E = \frac{512^2}{p^2} \cdot c$, influence reconstruction metrics such as FID, IS, SSIM, and PSNR. $E$ shows trends similar to 256p results (Figure 3). However, achieving comparable rPSNR/rSSIM to 256p requires $4 \times E$ for 512p reconstruction, which indicates that compression ratio of pixels to channels should be fixed to maintain performance.

## A.3. GAN Fine-tuning Ablation

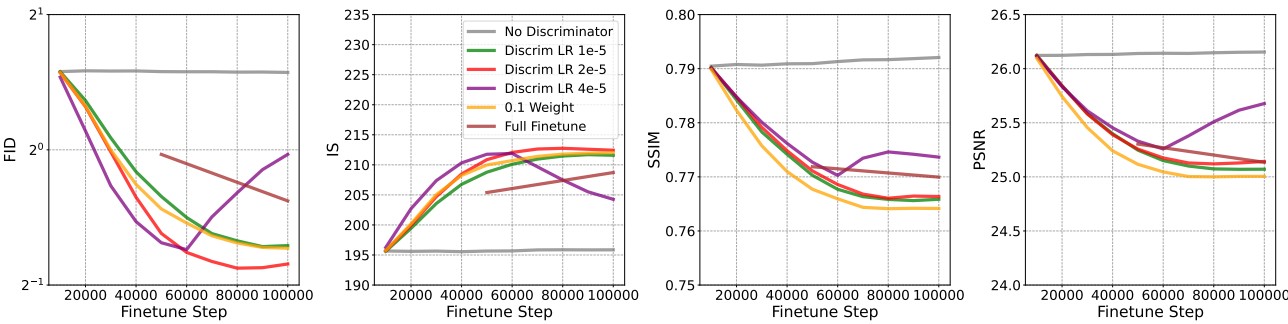

*Figure 13.* **Finetuning the Decoder with a GAN.** We study the effects of finetuning the decoder in ViTok S-B/16 on 256p images. We compare: (1) no GAN finetuning, (2) different discriminator learning rates, (3) an increased GAN loss weight (0.1), and (4) a full finetuning of all model parameters (including the encoder). The best results occur with a discriminator learning rate of $2 \times 10^{-5}$, while higher rates cause instabilities. We also observe a clear shift toward more generative behavior: as the decoder gains better IS/FID, it sacrifices some SSIM/PSNR, reflecting its transition into a stronger generative component.

In Figure 13, we study how various loss settings affect finetuning of the GAN decoder. Our goal is to highlight the trade-off and the transition of the decoder to more generative behavior. We use ViTok S-B/16 on 256p images, following the protocol in Section 2.3 for stage 2 fine-tuning from a model trained on stage 1.

We compare:

- Finetuning the decoder with the same Stage 1 loss (no GAN).

- Finetuning with discriminator learning rates ($\{1 \times 10^{-5}, 2 \times 10^{-5}, 4 \times 10^{-5}\}$) and a GAN weight of 0.05.

- Finetuning the full encoder/decoder with the GAN.

- Using a higher GAN weight of 0.1 with a discriminator learning rate of $1 \times 10^{-5}$.

From Figure 13, the best setting is a GAN weight of 0.05 and a discriminator learning rate of $2 \times 10^{-5}$. A higher discriminator learning rate causes training instabilities, while a lower rate degrades performance. Full fine-tuning produces results, but slightly underperforms when compared to fine-tuning only the decoder. Fine-tuning without a GAN yields no improvement, confirming that GAN training is the key factor behind the better results.

Finally, we see an inherent trade-off: improving FID tends to worsen SSIM/PSNR, indicating that as the decoder focuses on visual fidelity, it shifts more toward generative outputs. This demonstrates the evolving role of the decoder as a generative model to improve visual performance.

### A.4. Latent ViTok and Masked ViTok

In this section, we describe two variants of ViTok that provide different potential directions for tokenization. First we describe and evaluate our latent variation that does 1D tokenization and can form more arbitrary code shapes, then we discuss and evaluate our masking variant that allows for variable, adaptive tokenization.

**Latent ViTok Variation.** Another variant of ViTok involves utilizing latent codes following Titok (Yu et al., 2024). Initially, after applying a tubelet embedding, we concatenate a set of 1D sincos initialized latent tokens with dimensions $l_{\text{latent}} \times C_f$ to the tubelet token sequence $X_{\text{embed}}$. This combined sequence is then processed through the encoder and bottleneck using a linear layer. Subsequently, the tubelet tokens are discarded, and the latent tokens generated by the encoder form $Z = l_{\text{latent}} \times 2c$, from which we sample $z \sim Z$. This gives us a 1D code with easy shape manipulation since $L$ and $c$ are arbitrarily decided and do not depend on $p$. In the decoder, $z$ is upsampled to $C_g$, and we concatenate a flattened masked token sequence of length $L \times C_g$ with the upsampled latent code $l_{\text{latent}} \times C_g$. The decoder then predicts $\hat{X}$ in the same manner as the simple ViTok variation using masked tokens. This approach allows for a more adaptive compression size and shape using self-attention. In addition, it accommodates arbitrary code shapes of different lengths than $L$, provided there is redundancy in the code. A trade-off compared to the simple ViTok is the increased total sequence length and computational cost (FLOPs) during encoding and decoding. We refer to this variant as Latent ViTok.

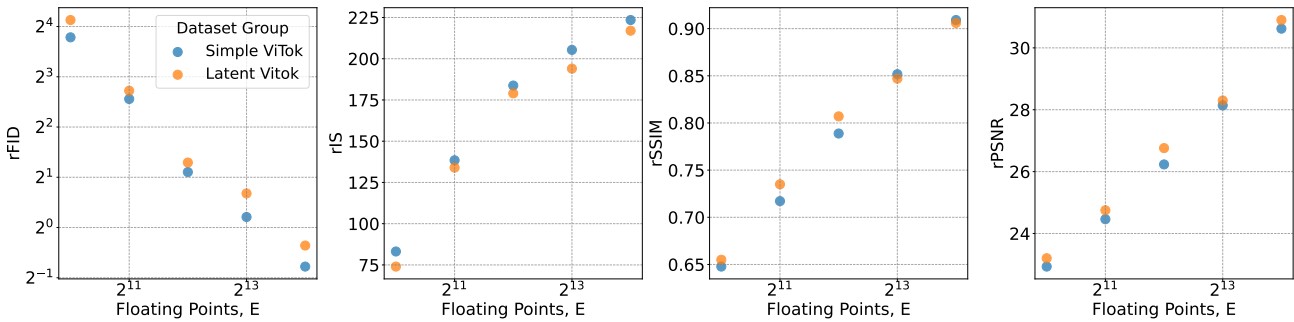

*Figure 14.* **256p Simple vs Latent ViTok Results.** We implement a latent variant of ViTok S-B/16, with $p = 16$ and $L \in \{64, 128, 256, 512, 1024\}$ latent tokens appended to the original patch embedding, then processed using full self-attention, and subsequently bottlenecked to $c = 16$. Although this latent variant slightly underperforms the simpler version in rFID/rIS, it remains comparable overall and follows the same rules as $E$. Consequently, it provides an alternative to Simple ViTok with greater control over the latent space.

We train latent ViTok on stage 1 (Section 2.3) where we fix $c = 16$ and sweep the number of latent tokens $L \in \{64, 128, 256, 512, 1024\}$ to adjust $E$. The results are shown in Figure 14. Our simple variant outperforms the latent version for most values of $E$, although the latent version achieves slightly better rSSIM/rPSNR for certain choices of $E$. This indicates that the latent approach is a promising alternative to simple ViTok for more control over the latent space; however, it comes with an increased computational cost due to the longer sequence of concatenated tokens. We leave this implementation out of ViTok due to added complexity.

**Token Compression via Random Masking.** The simplest bottlenecking process in ViTok involves manipulating $c$, which does not compress the number of tokens; the token count remains equivalent to the number tokens post-patching ($L$) or equivalent to the number of latent tokens ($l_{\text{latent}}$). However, manipulating $p$ does not provide a fine grain control over the token count.

To form another bottleneck, we can instead manipulate the main sequence of patch tokens by masking a random power of two number of tokens, starting with tokens *at the end* of the sequence and masking towards the beginning. This is similar to the method developed in ElasticTok (Yan et al., 2024). Matryoshka representation learning (Kusupati et al., 2024) encodes coarse-to-fine information in a single vector, letting any prefix serve as a valid embedding and yielding up to $14\times$

compression with no loss in ImageNet accuracy. Learned ordered representations with nested dropout (Rippel et al., 2014) train networks to rank latent dimensions by importance, so early codes cover low-budget tasks while later codes refine detail. Adaptive length tokenization via recurrent allocation (Duggal et al., 2024) iteratively refines image patches, producing a variable 32–256 1-D tokens that scale with entropy and task difficulty. Finally, Visual Lexicon (Wang et al., 2024b) maps images into the language token space, retaining fine visual detail in just a few tokens that can be mixed seamlessly with ordinary text prompts. For example, if we randomly select 256 as the masking amount for a sequence of 1024 tokens, then the last 256 tokens will be masked out and replaced with a learned masked token of dimension $c$. This directional masking strategy enforces an ordered structure to the tokens. We set the minimum length to $l$. The length of the code at inference, $l_{\text{eval}}$, forms another axis to change the code shape (Section 3) and $E = l_{\text{eval}} \times 2c$.

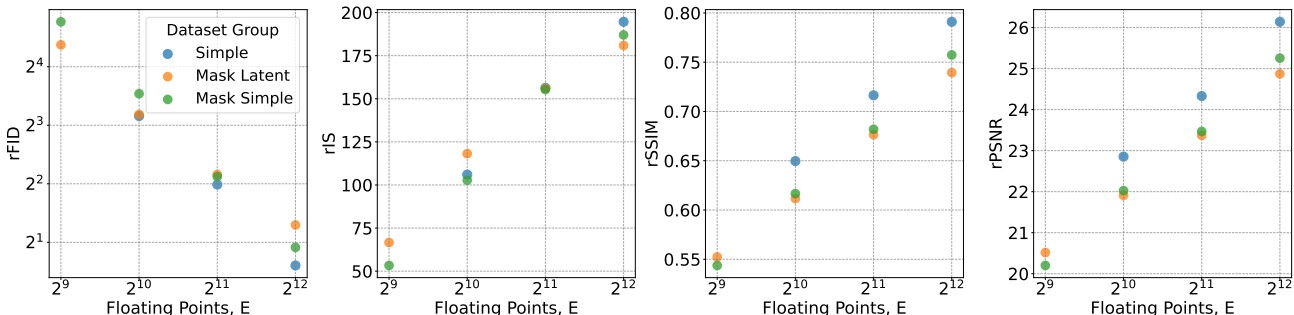

*Figure 15.* **256p Adaptive Masking ViTok Results.** We investigate variations of ViTok S-B/16 that apply token masking after encoding. We consider two approaches: *Mask Simple*, which masks the patch tokens following encoding, and *Mask Latent*, which introduces latent tokens (like the architecture used for Figure 14) and masks them. At stage 1 training time we randomly selected token lengths $\{32, 64, 128, 256\}$ with $c = 16$, then at inference evaluate each model on every token length and compare to the simple ViTok baseline at similar $E$. While the masking variations underperform the simple variant, they still perform strongly. *Mask Simple* tends to perform better at higher $E$, while *Mask Latent* achieves better results at lower $E$.

We now train our mask ViTok on stage 1 (Section 2.3) and investigate potential adaptive tokenization schemes. We first apply this masking strategy to the simple version of ViTok, directly masking the patch tokens after they have been processed by the encoder. We then explore the same approach on the latent version of ViTok. Both methods are trained with token lengths $\{32, 64, 128, 256\}$ and $c = 16$ on ViTok S-B/16 using 256p images.

Figure 15 compares these masking methods with simple ViTok across different $E$. While all masking variants slightly underperform the simple ViTok, their overall performance remains strong. In particular, masking patches directly is more effective for higher $E > 4096$, whereas masking latent tokens performs better when $E < 4096$. These findings highlight how ViTok can be adapted for flexible token lengths during inference, and illustrate how our method can be extended to learn an ordered structure of tokens. Though more work here is needed to improve performance further.

## B. Visualizations

In this section, we provide extra visualizations of generation examples from our various models and sweeps.

### B.1. Image Generation

Here provide generation examples from our final models and sweep conducted in Figure 4. The $p = 16$ visuals are shown in Figure 18, the $p = 32$ visuals are shown in Figure 19, and the $p = 8$ visuals can be found in Figure 20.

**Final ViTok model generations.** We provide example generations from the DiT models trained in Table 5. Visualizations are shown in Figure 16 and Figure 17.

### B.2. Video Generations

We include more video generation results in this section from Table 6 and show example generations at 1024 tokens in Figure 21.

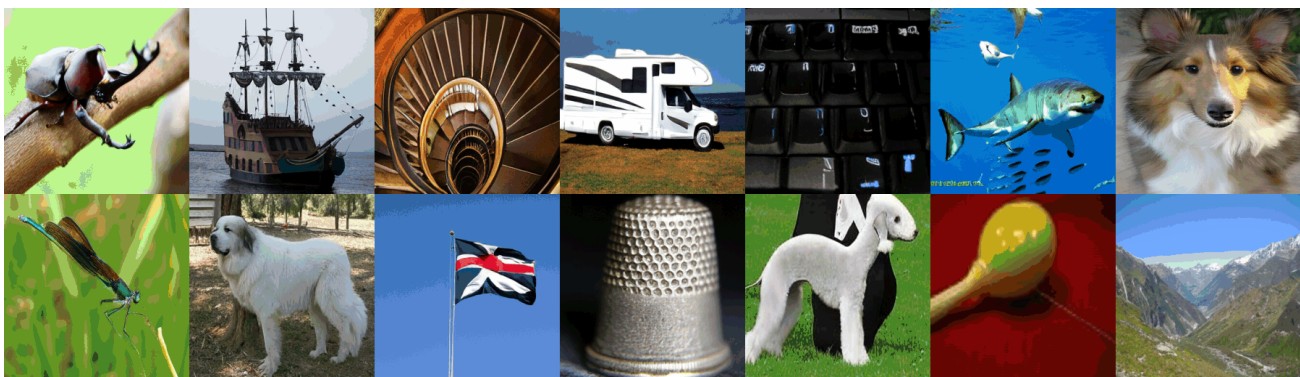

*Figure 16.* **256p image generation examples.** We show randomly selected 256p image generation examples from our DiT-XL trained using the ViTok S-B/16 variant for 4 million steps at a batch size of 256. Images were sampled with 250 steps using the DDIM sampler and a CFG weight of 4.0.

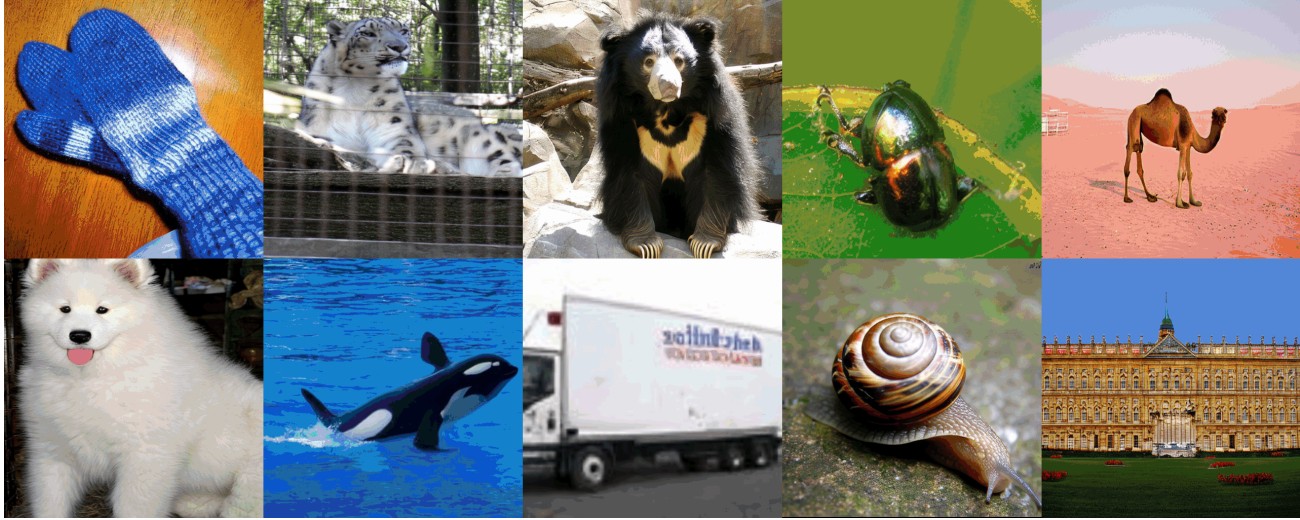

*Figure 17.* **512p image generation examples.** We show randomly selected 512p image generation examples from our DiT-XL trained using the ViTok S-B/16 variant for 4 million steps at a batch size of 256. Images were sampled with 250 steps using the DDIM sampler and a CFG weight of 4.0.

Patch Size 8, Channel 4

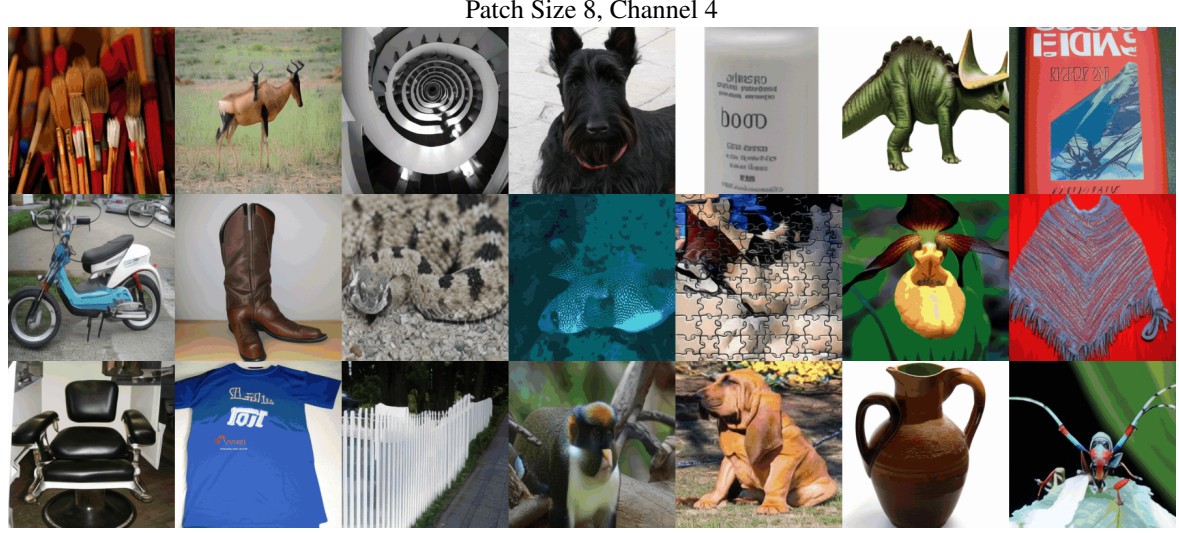

Patch Size 8, Channel 16

Patch Size 8, Channel 64

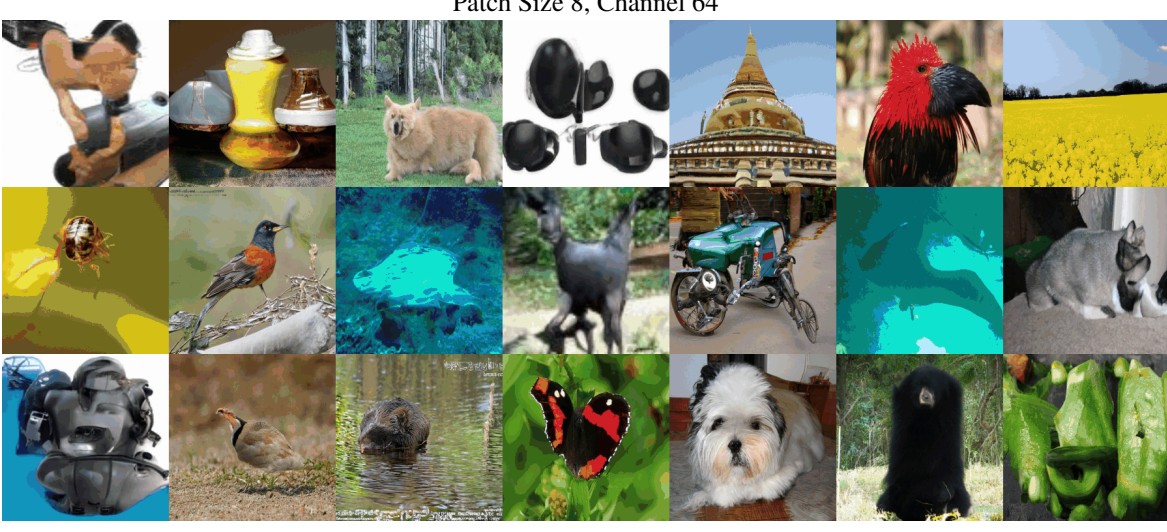

*Figure 18.* **Channel size generation visualization 256p for** $p = 8$**.** We show example generations for various compression ratios on ViTok S-B/8 from Figure 4. Here $c = 4$ has the best visuals that look close to good images, while $c = 16$ generally looks good as well but not as good. $c = 64$ looks very poor and the images do not look realistic.

Patch Size 16, Channel 4

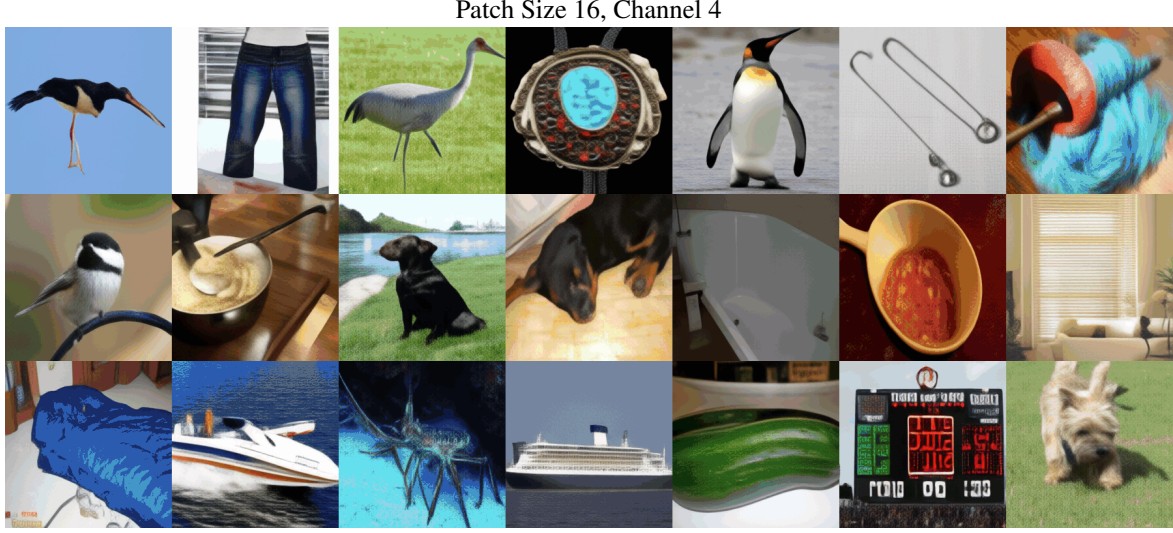

Patch Size 16, Channel 16

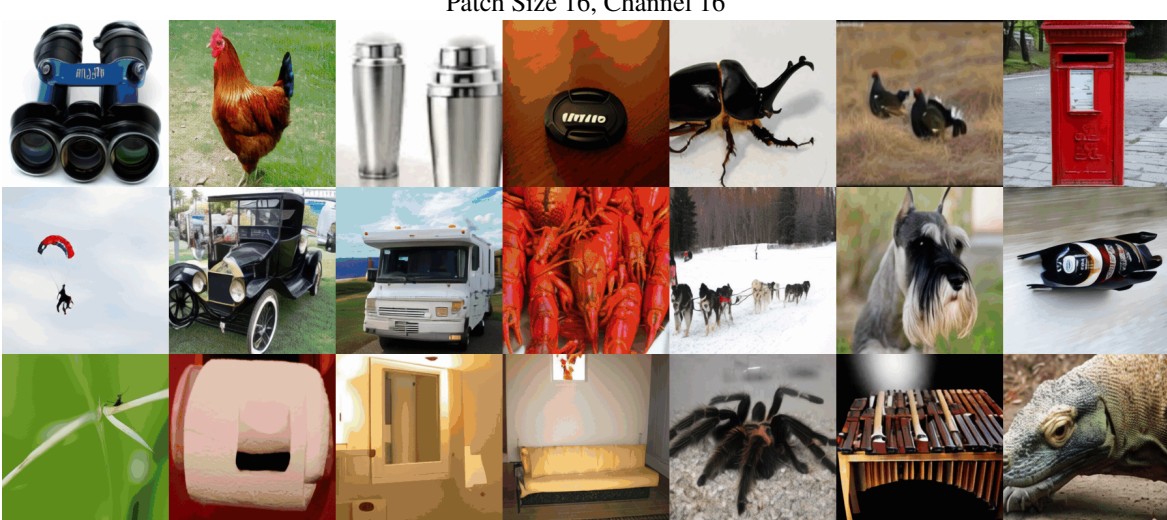

Patch Size 16, Channel 64

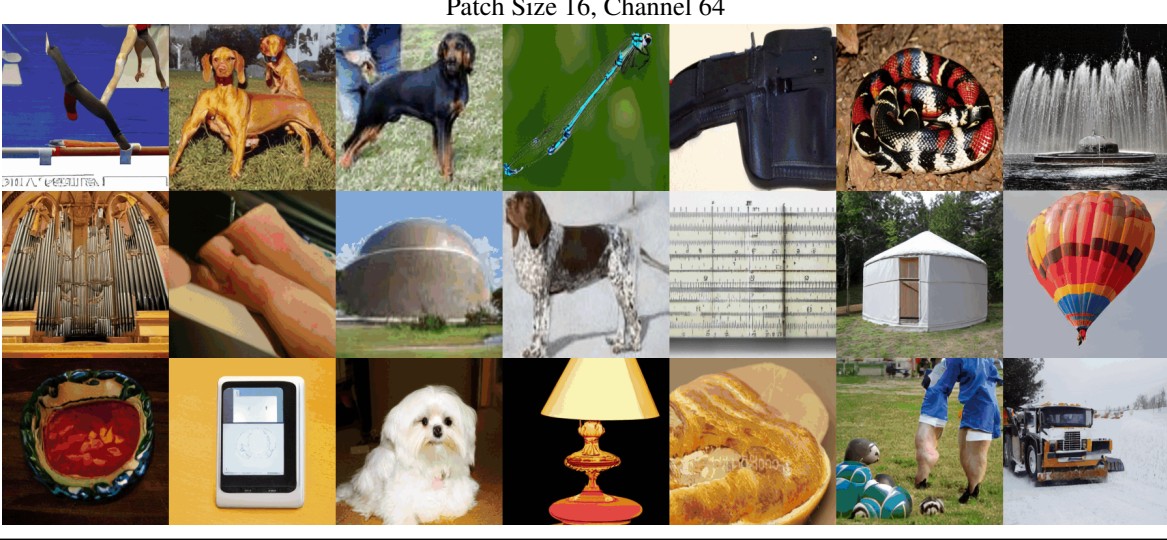

*Figure 19.* **Channel size generation visualization 256p for** $p = 16$**.** We show example generations for various compression ratios on ViTok S-B/16 from Figure 4 Here $c = 16$ has the best visuals that look close to good images, while $c = 64$ suffers artifacts that worsen image quality. $c = 4$ suffers from poor reconstruction quality from the auto-encoder.

Patch Size 32, Channel 4

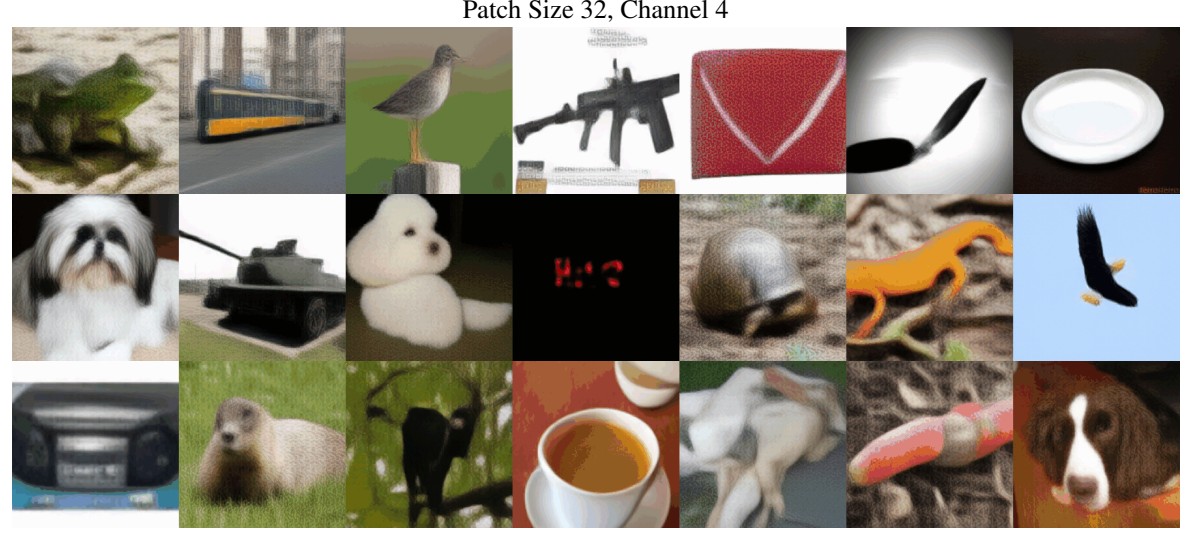

Patch Size 32, Channel 16

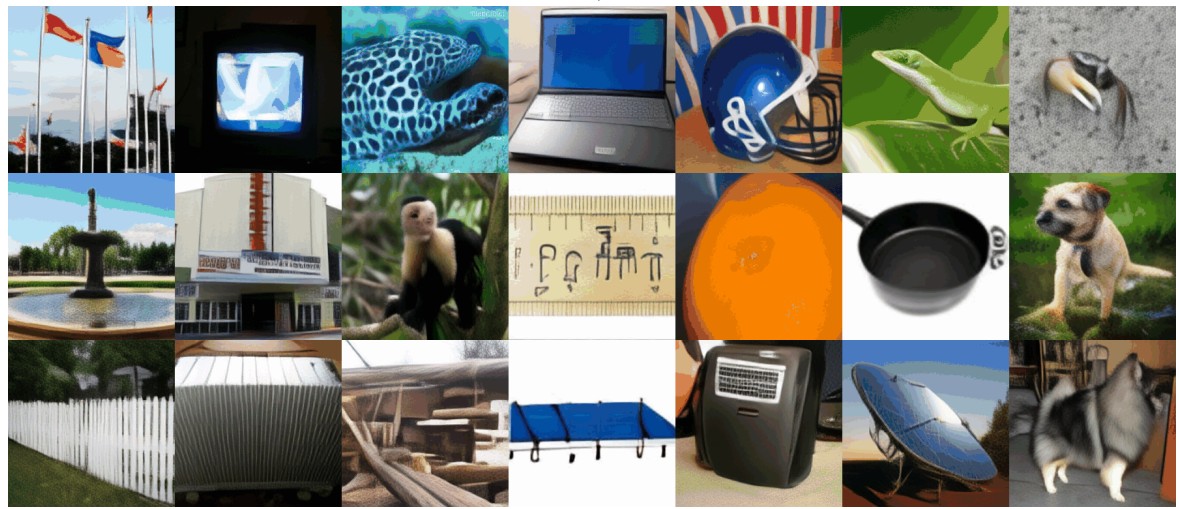

Patch Size 32, Channel 64

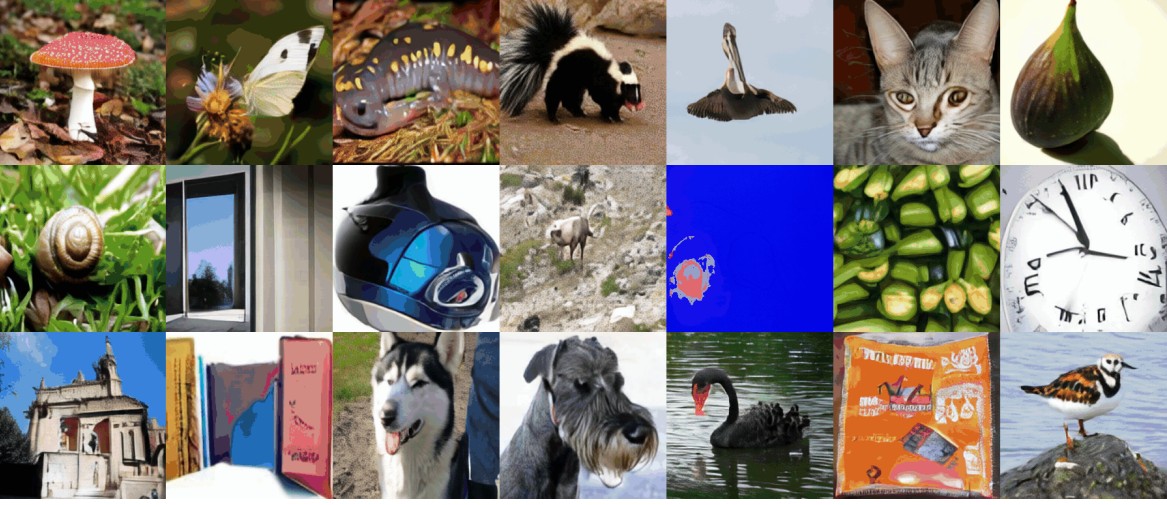

*Figure 20.* **Channel size generation visualization 256p for** $p = 32$**.** We show example generations for various compression ratios on ViTok S-B/32 from Figure 4. Here $c = 64$ has the best visuals overall but the high channel sizes make the image quality look poor and jumbled. Both $c - 16$ and $c = 4$ suffers from poor reconstruction quality from the auto-encoder.

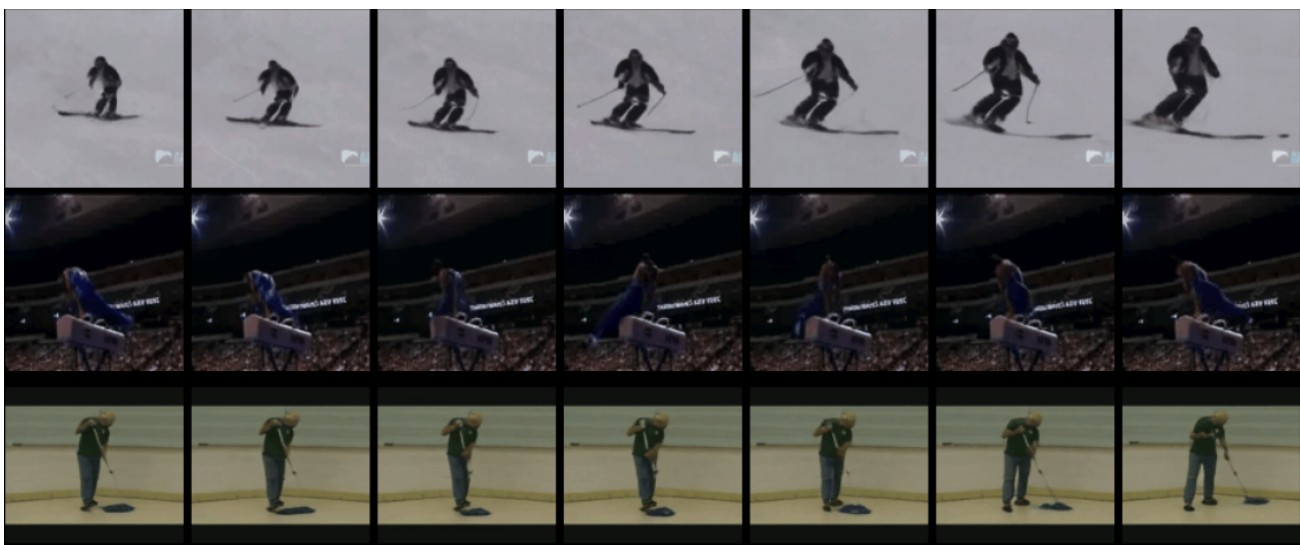

*Figure 21.* **128p video generation examples.** We show randomly selected 16×128×128 video generation examples from our DiT-L trained with ViTok S-B/4x8 variant. Videos are sampled with 250 steps and a CFG weight of 2.0.

