# OpenReview forum: "Learnings from Scaling Visual Tokenizers for Reconstruction and Generation"
_ICML.cc/2025/Conference — ICML 2025 poster_

### Official Review · Reviewer_QXLn · 2025-03-12

**Overall Recommendation:** 3

**Summary:**

Most current-day generative models operate on abstract representations rather than pixels, which are often produced by a VAE-like "stage 1" model. These models are commonly trained separately from the subsequent generative modeling step ("stage 2"), such as diffusion modeling. The central question the authors tackle in this submission revolve around scaling three aspects of VAEs, namely the encoder and decoder capacities, as well as the bottleneck size. The paper demonstrates that the most effective way to improve reconstruction performance is to increase the bottleneck size, instead of scaling encoder or decoder. When it comes to generation, there is a sweet-spot to that; after a certain point generation performance suffers with too expressive latents. The paper further demonstrates how scaling the encoder is not beneficial, while scaling the decoder can slightly improve reconstruction fidelity. Overall, the paper presents a "negative" result in that scaling VAE encoder and decoder size does not significantly improve downstream generative performance.

## update after rebuttal
The authors did not submit a rebuttal, which leaves all my questions and concerns open. Due to the uncertainty about those parts I lower my rating, but I still think the submission is overall a positive contribution.

**Claims And Evidence:**

Most claims are supported by enough evidence. There are, however, a few areas I would like to point out:
1. In several places (e.g. L333 "outperforming other CNN-based continuous tokenizers while utilizing significantly fewer FLOPs.", or L373 "achieves state-of-the-art (SOTA) rFVD results compared to other CNN-based continuous tokenizers with the same total compression ratio.") the paper suggests that ViT-based autoencoders are better than CNN-based ones. These claims are made in comparison to other models, trained on other datasets, with different training schedules and objectives. To support these suggestions, I'd like to see more controlled experiments; especially since the paper highlights the benefits of Llama's architecture modifications over more vanilla Transformers.
2. Section 3.2 and Finding 2: The authors claim that high E complicates convergence and degrades generation performance. This is demonstrated empirically in Fig 4 for a fixed stage 2 model size. That said, this result ignores the effect stage 2 model size / compute has on the optimal stage 1 compression ratio. Indeed, Esser et al. [1] demonstrate that larger stage 2 models benefit from less stage 1 compression. Stage 2 is usually the one that is scaled, which makes it important for explorations into stage 1 to be done with that in mind.

[1] Scaling Rectified Flow Transformers for High-Resolution Image Synthesis, Esser et al. 2024

**Essential References Not Discussed:**

1. Recently, there has been a focus on training tokenizers jointly with a regularization objective to make the latents more predictable by a downstream generative model, e.g. see [2,3]. These works show an alternative view on tokenization, that differs from the strictly separated stage 1 vs stage 2 view adopted in this paper. I would suggest the authors to add a discussion on these kinds of works and how it relates to their findings.
2. The appendix shows preliminary experiments towards 1D token representations, as well as ordered token sequences. I would suggest the authors to discuss the following relevant works: [4-7]

[2] When Worse is Better: Navigating the compression-generation tradeoff in visual tokenization, Ramanujan et al. 2024

[3] JetFormer: An Autoregressive Generative Model of Raw Images and Text, Tschannen et al. 2024

[4] Matryoshka representation learning, Kusupati et al. 2022

[5] Learning ordered representations with nested dropout, Rippel et al. 2014

[6] Adaptive Length Image Tokenization via Recurrent Allocation, Duggal et al. 2024

[7] Visual Lexicon: Rich Image Features in Language Space, Wang et al. 2024

**Experimental Designs Or Analyses:**

The experimental design and analyses appear sound. The authors provide enough implementation details that I would feel comfortable reimplementing the experiments.

**Methods And Evaluation Criteria:**

The evaluation criteria make sense to me. The authors evaluate on two common domains, image and video generation, and evaluate common metrics. I also am glad to see that the authors analyze per-image reconstruction metrics, and not only FID.

**Other Comments Or Suggestions:**

1. L143, Sec 3.1. Missing word in third line of paragraph, "For , ..."
2. Nitpick: The compression ratio in Sec 3.1 also depends on the precision of the latents and inputs.
3. L438: "workk"
4. Tables 2, 3, 4: I suggest showing the compression ratios (both token count and number of channels) of different tokenizers, along the Params and GFLOPs. As also noted by the paper, the compression ratio is one of the most significant hyperparameters that determines the final reconstruction performance; more so than encoder and decoder capacity (and presumably architecture). To illustrate the necessity for that, one could list a simple reshaping operation that performs 16x spatial compression at the cost of 16x the number of channels as a baseline with 0 Params, 0 GFLOPs, and perfect reconstruction metrics. These numbers can arbitrarily be gamed if the compression ratio is not shown.
5. Table 2: Why are the Params and GFLOPs of SDXL-VAE, OAI, and COSMOS-CI not listed?
6. Figures 16-20: The images are very compressed, and show a lot of banding. It's hard to evaluate some aspects of the tokenizer quality visually to to that strong compression.

**Other Strengths And Weaknesses:**

The authors investigate an understudied problem and show a clear negative result about scaling tokenizer capacity. This is a useful finding to the community, and may help reduce wasteful experiments in this area. It is also commendable that the tokenizers are not trained in a data-constrained setting, such as ImageNet, which would make me question some of the results. Overall the paper is well written and presented.

To me the main limitations of the submission lie in that stage 2 scaling is not taken into account, which may distort some of the findings. For example, it has been shown that scaled-up stage 2 models can take advantage of more expressive latent spaces, yet the experiments in this paper would suggest that the stage 1 bottleneck size should not be scaled beyond a certain threshold. Unlike for the bottleneck size, I am not aware of other papers investigating the impact of stage 1 encoder and decoder capacity when scaling stage 2 models, which makes me slightly less confident about that part of the results.

**Questions For Authors:**

For Section 3.2 / Figure 4, did the authors run any stage 2 scaling experiments? How does the U-shaped curve shift for larger stage 2 models / comptue?

**Relation To Broader Scientific Literature:**

The submission is timely, as there is a recent uptick in investigations into the role of tokenizers for downstream generative modeling tasks. Especially in the diffusion model literature, the effect of the first stage autoencoders has been understudied. Several tokenizer-focused papers, such as ViT-VQGAN or TiTok, investigate the stage 1 tokenizer scaling to some extent, but don't make it the main focus.

**Theoretical Claims:**

The paper does not make theoretical claims. All claims are empirical.

---

### Official Review · Reviewer_ocDZ · 2025-03-13

**Overall Recommendation:** 3

**Summary:**

This paper conducts empirical studies on training Variational Autoencoders (VAEs) for latent diffusion models using Vision Transformers (ViT) as backbones instead of commonly used CNN-based models. It reports two main findings: (1) The reconstruction quality of the VAE does not necessarily determine the generative performance of the subsequent diffusion model; (2) Scaling the encoder size contributes minimally to reconstruction quality, whereas scaling the decoder size substantially improves performance, suggesting the decoder functions primarily as a generator.

**Claims And Evidence:**

The paper’s claims are generally supported by comprehensive experimental evidence. Particularly, the extensive experiments effectively illustrate the authors’ main points regarding the relationship between VAE reconstruction quality, model scaling (encoder vs. decoder), and the generative performance of the diffusion model. However, the manuscript lacks depth in explaining why scaling the decoder significantly boosts reconstruction but does not substantially benefit the subsequent generative phase.

**Essential References Not Discussed:**

I didn't find there were essential reference missed

**Experimental Designs Or Analyses:**

Experimental designs are robust and extensive, effectively validating the primary empirical claims. However, certain important analyses are missing. Specifically, the authors fail to deeply analyze and explain the following critical points:
- How to optimally choose patch size and latent dimensionality given a fixed total floating-point budget (E)? While the authors suggest floating-point numbers in the latent codes are a bottleneck, the implications of different patch sizes on the complexity and generative quality remain unclear. Larger patch sizes would reduce token count and improve inference speed, but it is not evident from the current experiments whether this would degrade generation quality, increase model fitting difficulty, and why this occurs.
- Why does scaling the decoder improve reconstruction significantly but yield limited benefits for the second-stage generative performance? The paper lacks an in-depth theoretical or analytical exploration of this phenomenon.

**Methods And Evaluation Criteria:**

The methods and evaluation criteria chosen by the authors are appropriate for the research objectives. The choice of ViT-based VAEs and comparisons regarding encoder and decoder scaling are logical for investigating these questions. Nevertheless, the paper notably lacks sufficient visual demonstrations of reconstruction quality, which are crucial for applications dependent on accurate latent representations, such as conditional generation tasks.

**Other Comments Or Suggestions:**

The paper could significantly improve by providing more visual results for reconstructions, deeper theoretical or analytical insights, and explicitly addressing the trade-offs regarding latent dimension, patch size, and inference complexity.

**Other Strengths And Weaknesses:**

Strengths:
- Solid and extensive experimental validation clearly supporting the empirical claims.

Weaknesses:
- The results feel somewhat incremental, providing little novel insight beyond empirical verification.
- Lack of deep theoretical analysis or explanation of the observed phenomena.
- Insufficient visual evidence provided for reconstruction quality.
- Limited applicability of the ViT-based VAEs to a fixed image resolution.

**Questions For Authors:**

1. Given that floating-point numbers in the latent codes (E) represent a key bottleneck, how should one optimally select the patch size and latent dimensionality for the generative model? Specifically, would larger patch sizes, which significantly reduce inference complexity, negatively affect generative performance, and why? Have you conducted experiments or theoretical analyses addressing this issue?
2. While scaling the decoder substantially enhances reconstruction quality, why does this improvement not notably translate into better generative performance in the second stage? Can you provide deeper theoretical or analytical insights into this observation?
3. The paper provides limited visual demonstrations (only Figure 2) of reconstruction quality. Could the authors provide more comprehensive visual evidence? This is especially important for applications relying heavily on accurate latent representations.

**Relation To Broader Scientific Literature:**

The paper’s contributions mainly involve verifying and empirically supporting insights that are already widely accepted among researchers working with generative models, though not formally documented in existing literature. Its results clarify the roles of encoder and decoder scaling in ViT-based VAEs within latent diffusion models, contributing valuable experimental evidence for future reference.

**Theoretical Claims:**

The paper does not explicitly provide theoretical proofs or extensive theoretical justifications. Most claims are empirical and based solely on experimental results.

---

### Official Review · Reviewer_ZzLx · 2025-03-14

**Overall Recommendation:** 3

**Summary:**

This paper conducts an empirical study on scaling up the autoencoder for generative models, with a ViT-style autoencoder. The authors investigate bottleneck size, encoder size, and decoder size, and find the generation performance is not significantly better with scaling up the autoencoder. As a result, the authors also provide a setting of the autoencoder that achieves state-of-the-art performance.

**Claims And Evidence:**

Claims are supported by experiments. While some claims may sound too strong if not with the corresponding context. For example, L28 "when the bottleneck becomes large, generative performance declines due to increased channel sizes.", is not always true if without context (increasing the bottleneck channel from 4 to 8 sometimes can improve the generation performance).

**Essential References Not Discussed:**

I didn't find key missing related work.

**Experimental Designs Or Analyses:**

Experimental designs and analyses are generally good. One potential issue is that, when changing the design choice in the autoencoder, the way to train the latent diffusion model is fixed and classifier-free guidance (CFG) is also fixed. It is a reasonable way for ablation, while changing the autoencoder may potentially change the property of the latent space and thus change the optimal setting for training latent diffusion, for example, optimal CFG value may change.

**Methods And Evaluation Criteria:**

Method and evaluation makes sense. It is better to include human evaluation but it is too expensive given the large amount of experiments.

**Other Comments Or Suggestions:**

While I believe the findings are valuable, "scaling the auto-encoder alone does not significantly improve generative performance" as the key learning of this paper is not a very rigorous conclusion to me, as there are potentially other ways to scale up.

**Other Strengths And Weaknesses:**

The findings in this paper could be very helpful to the field for understanding and improving the current autoencoders.

While it could make this work stronger if having more interesting and deeper insights behind these findings.

**Questions For Authors:**

Is there any hypothesis about why scaling up the autoencoder does not significantly help generation? Do these findings give some hint on how to properly scale up the autoencoder, or do they just suggest that scaling up the autoencoder is not a promising direction?

**Relation To Broader Scientific Literature:**

This paper conduct comprehensive experiments on scaling up the existing autoencoder based on GAN, LPIPS, L1 losses for helping understand the properties of the autoencoders.

**Theoretical Claims:**

The paper is not about theoretical proofs.

---

### Official Review · Reviewer_mrYe · 2025-03-14

**Overall Recommendation:** 3

**Summary:**

This paper introduces ViTok, a Vision Transformer-based autoencoder, and systematically investigates the impact of scaling bottleneck size, encoder, and decoder architectures on image and video reconstruction and generation performance. Key findings reveal that the total latent floating points E is the dominant factor for reconstruction quality: scaling E enhances reconstruction capabilities but does not continuously improve generation performance. Meanwhile, excessive increases in channel size c degrade generation quality due to rising complexity in the latent space. Experimental results demonstrate architectural scaling asymmetry: encoder scaling yields no significant reconstruction gains and may even reduce generation performance, whereas decoder scaling improves reconstruction quality but offers limited marginal benefits for generation. By incorporating perceptual loss (LPIPS) and adversarial training (GAN), the model achieves flexible trade-offs between visual fidelity (SSIM/PSNR) and generation quality (FID/IS), enabling the decoder to serve dual roles in reconstruction and generation. Notably, video data exhibits higher compression efficiency due to temporal redundancy, achieving superior reconstruction quality at the same E compared to images. Experimental validation shows that ViTok achieves state-of-the-art performance on 256p/512p ImageNet-1K/COCO image tasks and 16-frame 128p UCF-101 video tasks, with computational costs (FLOPs) reduced by 2–5× compared to traditional CNN methods.

**Claims And Evidence:**

The paper systematically validates four core claims through experiments: the total latent floating points E is the decisive factor for reconstruction quality, with sweeps over patch size p and channel number c revealing a strong positive correlation between E and reconstruction metrics (rFID/rIS) (Figure 3), where scaling E significantly improves image/video structural similarity (SSIM/PSNR) but excessive channel size c degrades generation performance due to increased latent space complexity; encoder scaling shows weak negative correlations with generation metrics (gFID/gIS) (Figures 5, 7), indicating that expanding encoder depth/width yields no reconstruction gains and may harm generation quality; decoder scaling demonstrates strong positive correlations with reconstruction quality (rFID/rIS) (Figure 6) but limited marginal improvements for generation (Figure 8), highlighting its role in enhancing local texture details while contributing minimally to global generative capabilities; video data achieves superior reconstruction quality at the same E compared to images due to temporal redundancy, with experiments showing that rFID comparable to 256p images requires only 4–8× E instead of 16× direct pixel scaling (Figure 10), validating ViTok’s efficient exploitation of spatio-temporal redundancy. These findings provide theoretical foundations for visual tokenizer architecture design and reveal the inherent trade-offs between reconstruction and generation in current autoencoding paradigms.

**Essential References Not Discussed:**

Essential references have been discussed.

**Experimental Designs Or Analyses:**

Comprehensive ablation studies (e.g., varying p, c, and model size) and comparisons across datasets (ImageNet, COCO, UCF-101).

**Methods And Evaluation Criteria:**

ViTok uses 3D ViT with SwiGLU and RoPE for spatiotemporal modeling, trained on large-scale datasets (Shutterstock 450M images/30M videos). Standard metrics (FID, SSIM, PSNR, FVD) and ablation studies validate the bottleneck analysis. Generation is paired with DiT-L for downstream tasks. Rigorous ablation of architectural choices (encoder/decoder/bottleneck) and large-scale training.

**Other Comments Or Suggestions:**

None.

**Other Strengths And Weaknesses:**

To my understanding, this work demonstrates that scaling autoencoders does not effectively improve image generation performance. The paper conducts a detailed analysis of model scaling in terms of bottleneck size, encoder, and decoder architectures, showing that ViTok improves reconstruction capabilities. However, generation performance is much more important than reconstruction. How might the findings help researchers design better generators?

**Questions For Authors:**

Why in some experiments, better reconstruction performance does not lead to better generation?

**Relation To Broader Scientific Literature:**

This work discusses the effect of scaling autoencoders. It has little impact to other fields except image generation.

**Theoretical Claims:**

No formal theoretical proofs are provided. The paper focuses on empirical analysis of scaling trends.

---

### Decision · Program_Chairs · 2025-05-01

**Decision:**

Accept (poster)

**Comment:**

This paper analyzes the effects of scaling different parts of a ViT-based autoencoder on the performance of image/video reconstruction and generation. The authors did not provide any rebuttal. The final recommendations are 4 "Weak accept". The reviewers generally agree that
- experimental designs and analyses are appropriate and comprehensive,
- claims are supported by experiments, and
- findings are helpful for understanding and improving autoencoders (for reconstruction).

Their major concerns include
- only empirical results with no theoretical proofs or extensive theoretical justifications, and
- lack of positive directions on how to improve generation performance.

Taking both the above pros and cons into account, I recommend "Weak accept".